# Risk Management on Concrete Structures as a Tool for the Control of Construction Efficiency

Matej Špak, Tomáš Mandičák *, Marcela Spišáková and Dominik Verčimák

Faculty of Civil Engineering, Technical University of Košice, Vysokoškolská 4, 04200 Košice, Slovakia
* Correspondence: tomas.mandicak@tuke.sk; Tel.: +421-55-602-4378

**Abstract:** Risk management in the construction industry has a significant role in the impact of a construction project. Risk management is needed to make processes more efficient in terms of the implementation of construction projects. This research is oriented to the management of risks regarding the concrete structures of residential buildings in Central Europe. This research provides insights into risk management procedures, based on a risk's frequency, the probability of occurrence, and its impact on the effectiveness of construction projects for concrete structures. The aim of the research is to analyze the impact of risk management on the efficiency of construction projects for residential buildings and to propose a method of quantifying this impact, in terms of the technical and economic aspects, based on the calculated coefficients. This will assist in risk management by prioritizing those risks that will have the most significant impact on both the technical aspects and the efficiency of the project. The research results herein provide coefficients for determining the technical and economic impacts that can be implemented for risk management regarding the concrete structures used in the construction projects of residential buildings. The potential risks of concrete structures have been identified and divided into three groups. The scope and content of this study were chosen on the basis of the processes at individual stages. Construction management experts quantified the risk and probability levels according to the implemented projects. Based on the acquisition of these data, the rates of occurrence and impact, i.e., the extent of damaged parts (intensity), were determined. Subsequently, the significance factor was determined. To determine the efficiency, the net present value method was used, reflecting the investment's time value. The values were adjusted for time and inflation, affecting the overall efficiency coefficient of construction projects. These data were obtained from Eurostat.

**Keywords:** management and efficiency of construction projects; sustainability; net present value; risk factor significance

## 1. Introduction

The rapid development of society is connected with the rapid development of the architecture, engineering, and construction (AEC) sector. AEC contributes a crucial and vital part of the European Union (EU) economy, creating about 10% of the gross domestic product (GDP) in the EU and providing 20 million jobs [1]. At the same time, AEC is affected by a number of uncertainties and risks and presents a highly unpredictable environment [2] in comparison with other industries. The risks of a particular construction project are as unique as the construction projects themselves, along with their sources [3]. In order to eliminate the emergence of risk factors, it is necessary to create risk management (RM) policies.

However, risk management is important not only from the point of view of the fulfillment of some significant risks but also in terms of the efficiency of the processes of construction projects. According to some authors, efficiency means requiring the lowest possible costs and achieving profitability [4]. To achieve efficiency in construction projects, reducing costs is a priority from a financial point of view [5]. Risk management during the

construction of concrete structures should align with the construction project's efficiency. Therefore, the impacts of risk management can have an impact on performance indicators and, thus, on efficiency [6]. The simple idea is to reduce risk and be able to prioritize it in terms of efficiency. The goal is to find a methodology for calculating and prioritizing risks concerning the effects of efficiency in the construction industry.

### 1.1. Risks Analysis in Construction

Construction project risks can occur at every phase of a project, including design, implementation, operation, renovation, and demolition [7]. In general, the risk represents an unpredictable situation in the future that results in a negative consequence [8] or involves exposure to danger. In other words, risk means an exposure to loss or the likelihood of loss, multiplied by its probability of occurrence [9,10]. According to ISO standards, risk can be understood as a combination of an event's consequence and the probability of its occurrence [11]. Therefore, the risk can be expressed mathematically based on the function according to Formula (1):

$$R = f(P, C), \tag{1}$$

where $R$ is risk, $P$ is the possibility of a certain event occurring, and $C$ is the impact or consequence of the event on the final effect of that particular project.

The risks of a construction project mainly affect three basic construction parameters (price, time, and quality) [12].

#### 1.1.1. Risk Management Process

Risk management is a fundamental and essential matter for project managers. Inappropriately managed construction project risks can lead to the failure of construction projects [13]. Risk management involves the following:

- Risk analysis of a particular construction project and establishment of its procedures [14];
- Selection and classification of the particular methods for the risk management of projects [15,16];
- Application of the selected method or tool to a given matter within risk analysis [17].

Simply put, RM contains risk identification, risk analysis, benefit and cost analysis for variant estimation, and risk assessment and monitoring [18]. RM is established by the standard ISO 31000, which provides guidance on how to develop, implement, and continuously improve the RM process in all activities of the project [19]. The RM process consists of the following steps: establish the context, conduct a risk assessment, and enact risk treatment. All steps are permanently monitored [19,20]. Risk assessment includes the following:

- Risk identification;
- Risk analysis;
- Risk evaluation [19].

The key step in RM is risk identification. The process of risk identification can be achieved using different methods, techniques, and tools, which include the Delphi technique, interviews, brainstorming, presumption analysis and constraints, cause–effect diagram/analysis, document review, SWOT analysis, fault tree analysis, previously learned lessons, risk breakdown structure (RBS), and root cause analysis (RCA) [7,10,18].

The next step in RM is risk analysis and evaluation. Two basic approaches to risk analysis are generally known, namely,

- Qualitative;
- Quantitative [21].

The quantitative approach is primarily based on the probability of risk occurrence. This approach provides objective results in the case of sufficient and relevant data. On the other hand, the qualitative approach is often based on a subjective opinion, experience, or

an evaluation of a specific matter [22]. The methods that are used for risk identification (the estimation of probability/impact, brainstorming, a cause and effect diagram, expert judgment, checklists, the Delphi method, risk data quality assessment, risk breakdown matrix (RBM), event tree analysis (ETA), and a probability and impact matrix (PI matrix)) are very often also applied for qualitative analysis [7,10,18,23,24]. The core quantitative techniques are fault tree analysis (FTA), decision tree analysis, probability distribution, sensitivity analysis/tornado diagram, fuzzy logic, expected monetary value (EMV), analytic hierarchy process (AHP), and the Bayes network [7,10,18,20,21,23]. At present, Monte Carlo simulations and the application of system dynamics as computer-based simulations for analysis techniques are widely used.

Based on the results of the risk analysis and its evaluation, the resulting risk response plan provides reasonable actions for the elimination of construction project threats and increases project opportunities. The degree of severity is different. According to the severity of the risk, it is also necessary to apply the appropriate techniques and tools for risk response planning. Based on the nature of their use, we can apply the following techniques:

- Prevention risk management;
- Remedial risk management [9,25].

The most appropriate approach is risk prevention. Thus, it is necessary to use methods to prevent possible risks already or ensure their significant elimination in the design/planning phase of the construction project. Some of the risks of a construction project cannot be managed in the design phase of the project. It is impossible to predict them in advance. Then, the remedial method becomes important [10,26,27].

The selection of appropriate methods for the identification, analysis, and evaluation of risks depends on the size and type of the construction project, the financial implications, available information, the time available, the extent of innovation, the experience of the analysts, and expected results [22].

1.1.2. Classification of Construction Project Risks

Risks in construction projects can be classified according to different aspects. Based on the analyses made in many studies [10,28–38], we can distinguish three basic aspects of the types of construction risks:

- The type of relationship between risk and the construction process: direct and indirect risk [28];
- Construction project risks occurring in a specific phase: risk of design, construction, usage, maintenance, renovation, or demolition phases [10];
- Construction project risks influenced by a specific stakeholder: investor, designer, contractor, and subcontractor [29];
- Level of construction risk: project, region, and market levels [30];
- Risks of the particular parameters of the construction process: economic, technological, environmental, safety, and quality [31].

Construction project risks are directly or indirectly related to the construction process. External risks are not directly related to the implementation phase of the construction project, but they do affect its completion. Examples of these risks include environmental, political, economic, and legal risks. Internal risks are directly related to the construction process and can be planned, controlled, and managed by the construction project manager. They can be categorized as follows: design, technical, construction, management, and finance risks [28,32].

Many studies [33–36] have also addressed risk management in the context of sustainable construction. According to Erdenekhuu et al., sustainable construction represents the responsible construction and management of green buildings, including the use of renewable resources and the application of environmental principles [33,34]. At the same time, a sustainability risk framework, which supports the identification of risks in sustainable construction, was compiled [37]. Just as there is risk management for sustainable

construction, there are also risks related to sustainable construction, e.g., energy-saving uncertainty, pollution restrictions, and changes in green building policies [34,38].

### 1.2. Processes of Concrete Structure Buildings

In the construction practice of the European Union, the process of making concrete structures is governed by the conditions defined in the technical standard EN 13670 [39]. The process of constructing concrete buildings is directly linked to other processes that are necessary for the making of a concrete structure. It is mainly a process of formwork and reinforcement of the future structure. The entire concreting process can be divided into the following:

- Preparatory process;
- Subsidiary process;
- Transport process;
- Main process.

The preparatory process consists of concrete production and the testing of concrete. Production consists of the dosing and mixing of the individual components of concrete, such as cement, aggregate, water, admixtures, and additions. Concrete as a building material (according to Regulation 305/2011) must meet basic requirements and properties [40]. These properties are prescribed in the technical standard EN 206 [41]. To verify the required properties, concrete quality control is carried out at the production site through production control tests. A solid real-time risk management solution is a must to identify machinery inefficiencies and prevent breakdowns [42,43].

The subsidiary process consists of the production of formwork and the supplementary structures necessary for the production, transportation, processing, and treatment of fresh concrete, such as scaffolding and protective structures. Formwork, including supplementary structures, is an integral part of the production of monolithic concrete structures. Formwork can be temporary or permanent, which forms an integral part of the finished structure (lost formwork). The subsequent process is transportation, which can be divided into off-site transport (transportation from the place of production to the place of concrete processing) and construction site transport (transportation within the construction site to the place of concrete storage). Part of the transport process is also the transport of concrete components and the transport of the machines and equipment necessary for the preparatory, subsidiary, and main processes of concreting. The last process is the main process, which includes the processing of fresh concrete (placing, compaction, and surface treatment) and the treatment of placed concrete. The main process is the one most susceptible to malfunctions. In the main process, there are many procedures that are significantly influenced by the human factor.

During every process, there is a risk of not meeting the required standard of quality. Each output in the production of concrete structures is largely individual and specific due to the scope and content of each project. Therefore, the influence of various factors has a specific impact on the quality of the construction work. In every phase of construction, there is a certain risk potential that can affect the quality of the construction negatively. From the point of view of the sustainability of construction, it is necessary to minimize these risks as much as possible. Therefore, a detailed analysis of all the processes that occur within the stages of implementation is necessary. Concrete structures are implemented in almost every construction project. In addition to their impact on the quality of construction, factories related to individual processes also have an impact on the time and cost efficiency of construction work. There are several methods for the risk assessment of construction [44–47], but each of them is specific to individual conditions.

### 1.3. The Impact of Risks on Construction Efficiency

The impact of risks on the efficiency of construction projects can be defined in several ways. The first step is to define "efficiency" in the construction project. According to this

study, efficiency is perceived as a view of the performance of construction projects; that is, added value and the resources used to achieve goals.

Several authors perceive performance and efficiency as key indicators of the overall performance of construction projects. Financial indicators are often considered among these key performance indicators of efficiency. On the one hand, there are revenues from sales or rent, profit, and profitability indicators. On the other hand, these are cost parameters. Therefore, construction project costs are among the most basic efficiency indicators. The construction time is closely related to these costs, as has been confirmed by several studies.

The fact that risk management, efficiency, and key performance indicators are often addressed is also shown in an extensive study that analyzed several studies and research on this topic [48]. The perception of costs as a primary financial parameter in the management of construction projects is a way of quantifying the project's results in an economic form. This is often the primary business goal of construction companies when implementing construction projects. Optimization, the effort required to obtain capital and improve the processes that will lead to increased performance, is needed in every construction project. According to [49], every construction company must work on process optimization in its procedures. It can only be competitive, prosperous, and break into new markets if it pays attention to this aspect. Optimization can increase resource efficiency, reduce construction time, and minimize construction costs.

In the context of the effectiveness and risk management of construction projects, the risks associated with procurement and payment processes, payment and delivery, and receipt processes are mainly addressed. Additionally, based on this study, indicators such as costs, profits, and construction project implementation time can be considered important. Individual failures could have a negative impact on these basic efficiency parameters. The jury of experts also addressed the issue of risk perception and individual factors concerning the effectiveness and performance of construction projects, trying to identify the primary factors and impacts on the construction project. Meanwhile, understanding how all stakeholders perceive specific criteria can significantly impact project outcomes [50]. For example, the authors of [51] investigated safety and risks on construction projects and reported that stakeholders can usually identify critical safety risks but have different risk probability estimates, creating some differences. The literature in this area states that many researchers have noted differences in risk perception among key stakeholders regarding safe construction work [52]. However, few have offered empirical data to support their claims [51].

Risk management in construction projects is an essential part of the life cycle cost management of a project; the project cost risk management system includes the identification of management objectives, risk factor analysis, risk identification, risk assessment, risk management, and risk information feedback [53].

Another study pointed to the impact of project delays as a risk that affects efficiency. The authors investigated the critical factors affecting contractors' efficiency in Indian construction projects [54].

It is important to note that efficiency largely depends on the successful implementation of projects [55]. Factors such as high costs, the complexity of construction projects, and an unfavorable economic environment reduce the probability of construction projects' success and increase the risk level [56,57]. At the same time, the opposite is also true, as the fulfillment of risks in a construction project increases costs and is one of the main parameters of the success of construction projects [58].

When estimating the degree of risk and probability, a valuation method based on the net present value of the project, or part of the construction works, will also be of use. For example, a study from Poland pointed out the suitability of this method in assessing the impact of risk on the performance and efficiency of projects [59].

The net present value of the efficiency of a construction project in random implementation conditions is a probabilistic indicator of the profitability and efficiency of construction projects [60]. It can be the basis not only for the decision to suspend or continue the prepa-

ration and implementation of the construction but also when evaluating the impact of the risk on the construction process. This indicator can be estimated reliably, reasonably, and comprehensibly, and its use is convenient and straightforward. It is a good measure of the net present value of the effectiveness of a project under random implementation conditions. Moreover, the profitability of the project is easy to interpret. Together with analyses of the risk of total costs and the risk of total returns, it is a comprehensive assessment of the project's viability and the impact of fulfilling the analyzed risks on the construction project [59].

McKinsey's study on infrastructure construction projects describes the steps leading to the management and evaluation of the impact of a particular risk on performance and efficiency [61]. The first step involves planning an acceptable risk strategy. The next step is the risk identification and quantification of the probability of occurrence. Economic modeling offers several advantages, based on which process it is possible to manage contracts effectively, improve the selection of suppliers, and streamline the setting of deadlines, along with other attributes that can significantly contribute to reducing the probability of project failure due to the fulfillment of some risk. Risk monitoring and setting up other decision-making processes will help to manage the project and its results better. Lastly, the calculation of value and impacts on effectiveness through net present value (primarily via parameters such as costs and sales over time, which have an impact on this indicator) are also important.

Based on the aforementioned studies, it is assumed that if the predicted risk is fulfilled, this has consequences for the economic efficiency of the construction project. Due to the fact that each technological process lasts a different length of time and also that the probability of risk and the costs associated with solving the problem are different, it is necessary to take these factors into account when finding a methodology for calculating the risk's impact on economic efficiency. For these purposes, the net present value appears to be the best solution, representing an indicator that takes into account all these aspects and that can also be expressed relatively.

Steps leading to the management and evaluation of the impact of risks on performance and efficiency are following:

- Planning and portfolio strategy;
- Determine single-project economics;
- Determine finance structure, debt ratio, ownership;
- Manage construction, budget, timeline, quality;
- Optimize net present value and performance for asset;
- Manage assets in portfolio [61].

## 2. Methods Applied

### 2.1. Research Goal

The construction of concrete structures involves a number of processes that can influence the performance of the structure significantly. The factors related to these processes also affect the efficiency of the construction, both from the point of view of time and the costs of construction. Our research goal was to analyze and verify the degree of impact of individual factors on construction efficiency during the building of concrete structures in the selected types of buildings. The impact of risk management analyses on the efficiency of construction projects for residential buildings is based on the calculated coefficients from a technological and financial perspective. This will help risk management by prioritizing the risks that have the most significant impact on the technical side and the efficiency of the project. The research results are important because we obtained coefficients for determining the technical and financial aspects that can be implemented for the needs of risk management on the concrete structures of residential buildings. Some risks can affect the construction processes more than others [2]; therefore, it is necessary to know the intensity of the technical and financial impacts on the efficiency of building projects.

### 2.2. Determining the Importance of Factors and Data Collection

To determine the possible influence of various factors on quality during the realization of concrete structures, selected risks were investigated. These risks were divided into three groups. The scope and content of the groups were designed according to the processes taking place in the individual stages of the production of concrete structures. The groups were divided into formwork (designated by F), reinforcement (designated by R), and concreting (designated by C). Individual risks were selected based on the analysis of already completed construction projects. Each group was subsequently divided into three parts, according to the time phase of the process implementation. The first phase represents the transport process (designated by D), the second phase is the preparatory process (designated by P), and the third phase is the main process (designated by R). The individual processes, classified according to the relevant groups (F, R, and C), are listed in Table 1.

**Table 1.** List of processes causing potential risks.

| Desig. | Formwork Group | Desig. | Reinforcement Group | Desig. | Concrete Group |
|---|---|---|---|---|---|
| F-D_01 | storage area for formwork | R-D_01 | storage area for reinforcement | C-D_01 | working area for vehicles |
| F-D_02 | documentation | R-D_02 | documentation | C-D_02 | quality control plan |
| F-D_03 | completeness of the set | R-D_03 | completeness | C-D_03 | control of delivery note |
| F-D_04 | completeness of elements | R-D_04 | completeness of accessories | C-D_04 | project/note conformity |
| F-D_05 | completeness of accessories | R-D_05 | marking of parts | C-D_05 | quality control of concrete |
| F-D_06 | marking of parts | R-D_06 | reinforcement: shape | C-D_06 | delivery schedule plan |
| F-D_07 | supporting: shape, integrity | R-D_07 | surface: cleanliness | C-D_07 | supplier communication |
| F-D_08 | panels: shape, integrity | - | | - | |
| F-D_09 | surface: flatness, integrity | - | | - | |
| F-D_10 | surface: cleanliness | - | | - | |
| F-P_01 | preparation area | R-P_01 | preparation area | C-P_01 | readiness of workers |
| F-P_02 | tool readiness | R-P_02 | tool readiness | C-P_02 | framework control |
| F-P_03 | availability of release agent | R-P_03 | availability of energy | C-P_03 | reinforcement control |
| F-P_04 | energy and water availab. | R-P_04 | storage area for structures | C-P_04 | project of concreting |
| F-P_05 | material for suppl. parts | R-P_05 | supplementary parts | C-P_05 | property of the equipment |
| F-P_06 | handling equipment | R-P_06 | handling equipment | C-P_06 | tool readiness |
| - | | - | | C-P_07 | preparation for treatment |
| F-R_01 | dimensions and shape | R-R_01 | type and dimension | C-R_01 | place of concrete loading |
| F-R_02 | position of formwork | R-R_02 | position of reinf. elements | C-R_02 | suitable method of loading |
| F-R_03 | stability and strength | R-R_03 | position of suppl. parts | C-R_03 | ensuring loading time |
| F-R_04 | strength, tightness of joints | R-R_04 | spatial stability | C-R_04 | cold joint compliance |
| F-R_05 | compliance with the project | R-R_05 | compliance with the project | C-R_05 | method of compaction |
| F-R_06 | position of installations | R-R_06 | cleanliness of reinforcement | C-R_06 | treatment of loaded concrete |
| F-R_07 | position of openings | - | | - | |
| F-R_08 | position of other elements | - | | - | |
| F-R_09 | release agent on surface | - | | - | |
| F-R_10 | access to the work platform | - | | - | |

The share of residential buildings in the total construction implemented in Central Europe is around 50%. Development companies or private investors implement most of these building projects. One of their main goals is to complete the project as quickly as possible while maintaining the required quality. This is why this research has focused on residential buildings. The data collection took place in 2022. A total of 12 contractors from Central Europe were contacted who constructed concrete structures in the aforementioned period, with 18 different projects taking place around the investigated area. Some large-scale projects (nos. 3, 7, 11, 15, and 16) provided data from several concreting operations. In total, the survey focused on 26 separate constructions of concrete structures.

The representatives of the approached contractors answered questions regarding the occurrence of risks (according to Table 1) in each group and for each project separately. The answers were processed using a binary system. Thus, if the given risk was relevant within

the project or to concreting, the value was "1". Otherwise, the value was "0". Each of the 67 processes in the individual groups (F, R, and C) and phases (D, P, and R) was evaluated for each project. In addition to the occurrence of risks, data regarding the actual impact of a specific risk on the given process were also obtained from the contractors. The risk impact value ranged from 0 to 100%. The contractors themselves determined the percentage impact of every relevant risk on the project costs, based on the actual increased costs.

The impact of the occurrence of risk on both the construction time and costs was also investigated. Data on time values were determined based on the standard labor needed for a specific process.

### 2.3. Data Processing

Data regarding individual constructions that were obtained from contractors were used for the determination of the degree of occurrence of a risk in a given process and the degree of influence of the occurrence of a risk on that process. The rate of occurrence of the risk represents the probability (P) and the rate of impact of the risk; in this case, it represents the intensity of the impact of the given risk (II). Subsequently, the significance factor (FS) was determined from the values, according to Equation (2).

$$FS = P * II \tag{2}$$

The perception of investments in the construction industry is mainly based on the costs associated with the construction project and the time scale in which the costs will be invested. However, this basic financial view also reflects the requirements and assessment of various investment options. It can also be applied to the fulfillment and impact of risks on the outcome of a construction project.

Research primarily works with cost data, which are not expressed as an absolute value, but in coefficients and ratios in relation to other processes. The advantage of this is that it applies to different project sizes. The coefficient adjusted for the time when the project will be moved also considers the time parameter and the impact on the value of the investment. Since the effects of the risks on the efficiency of a project are solved at the time of construction, sales, as an indicator of income from a given activity, are irrelevant at this stage. This parameter also takes into account the investment from the point of view of inflation. The result is again a coefficient that is adjusted for inflation and project delay due to risk fulfilment [62].

The net present value (NPV) formula was used for the calculation:

$$NPV = \sum_{t=0}^{T} \frac{C_t}{(1+r)^t} \tag{3}$$

where *C* is cash flow at time *t*, *r* is discount rate expressed as a decimal, *I* is time period.

Data were collected from the actual projects that were the subject of the research. Data related to inflation and time parameter modeling were based on Eurostat data.

To calculate the coefficient as a tool for creating a methodology and formula for general use and with applicability to various projects (except for the projects that are stated in Section 2.4), cash flow was used not as a nominal value but as a coefficient that can be used to represent any value in various projects. The cost item was expressed in the same way.

### 2.4. Research Limitations

This research focused on the risk management of concrete processes and was based on a large amount of experience from many projects in the field of residential building construction. Therefore, at this stage, it is necessary to state that the research focuses on this type of construction, and the results given in the form of the coefficient of the influence of selected risks on the effectiveness of construction projects are presented for this type of construction.

The first point for setting limits is the second building. These results for the given construction processes cannot be used for other types of construction; the results were only verified on residential buildings and construction projects.

Regarding future research, it is appropriate to focus on construction projects and research into other projects in the field of packaging infrastructure and road construction. Concrete processes form an equally significant part of this field and are the main element of construction.

Another limitation of the research was the fact that the research was carried out from the point of view of the climatic conditions in the area of Central Europe, which experiences typically mild weather and roughly the same conditions for all construction sites. However, they are applicable to other projects in other climate zones, for example, areas with a higher temperature or, conversely, with colder conditions. This may be the opposite of the assumed conditions that affect the speed of construction and the technological process of construction, which may lead to different durations of the selected processes. The time parameter was only taken into account when calculating the value and coefficient, so these results could be different in other climate bands. As this research was not extended beyond the conditions of Central Europe and the temperate zone, it reflects the types of buildings in this area.

Another research limitation may be the inclusion of the current value for the investment and analysis of the effort, in order to perceive the result and the effects on efficiency in the contexts of the time factor and of inflation. From a financial point of view, it is necessary to state that the research data are drawn not only from times when inflation was stable and oscillated for a long time at the level of 2%, but also during the period when there was a sharp increase in prices in the construction industry, and inflation reached a value of 15%. This factor is challenging to predict. Deviations may occur, even in economic analyses based on historical data. However, the advantages of the current research are that the results reflect not only an economically peaceful period but also a situation with an increase in prices in the construction industry.

## 3. Results and Discussion

The concrete structures achieved within the 26 monitored construction projects were of the same type, namely, reinforced concrete walls. For each individual process from each group, an average value from all construction projects was determined. This average value represents the probability (P) of the occurrence of the risk. The mean values ranged from 0.000 to 0.885. A zero value means that a given risk did not occur in any project. A higher average value means that more cases classified within the given risk occurred for all projects. A zero value was found only in the case of risk C-D_03 (control of the delivery note). This result confirms a theory deduced from many years of practical experience. The control of delivery notes during the delivery of concrete is essential not only for checking the quality and correctness of the delivery but also for the subsequent invoicing of the delivered concrete. Therefore, this factor is the least risky. Conversely, the highest values were found in the case of risks C-P_03 and C-P_04, these being reinforcement control before loading and the project of concreting, respectively. These two risks occurred most frequently in the survey. Checking the reinforcement before concreting is an often-neglected activity, but it represents the great risk of negatively affecting the quality of the concrete structure. Another risk with a high occurrence is the concreting project, which was used in only three cases (within just one construction project). The concreting project should be a standard part of the construction project documentation, but it is prepared and used only in exceptional cases. The standard deviation for individual processes was also determined. In most cases, the dispersion of values was relatively small. The maximum value of the standard deviation was 0.249 for risk C-R_03 (ensuring the loading time). The low values of the standard deviation confirm the relevance and small dispersion of the answers for the given risk.

The impact intensity determined by the contractors ranged from 0 to 100% (from 0.0 to 1.0). This value expresses an increase in costs and/or an extension of the time of the

construction process, due to the occurrence of a risk. Similar to the occurrence of risks, the average values for each risk were determined separately to ascertain the intensity of the impact. This average value represents the impact intensity (II). An increase of more than 30% was found in only 14 risks and was even more than 50% in 6 risks. The largest average increases were found in the reinforcement risk group and in the preparation and realization group. No increases were detected in the concreting risk group and in the preparation and realization group. The increase is expressed as a relative value, not as an absolute value. Therefore, it is not appropriate to compare the determined values for individual risks with each other.

The significance factor was determined from the probability and impact intensity for each risk separately. It expresses the level of risk impact, taking into account the rate of its occurrence in monitored construction projects. A zero value was found only in the case of risk C-D_03 (control of the delivery note). The maximum value was 47.93% for risk R-D_05 (the marking of parts) from the reinforcement risk group. Wrong or insufficient marking of a part of the reinforcement causes confusion and its use in the wrong place or the need for its replacement, which leads to a time shift in the implementation of the construction and increased costs. This risk is the most significant to be identified within the investigation and has the greatest impact on the technological parameters of the production of concrete structures (in Table 2).

**Table 2.** Final design of the factor significance (FS) and the impact of risks (from a technical and efficiency point of view) on construction efficiency during the building of concrete structures of the selected type of buildings.

| Factor | Impact Intensity (%) | Impact Intensity | Probability | Factor Significance (FS) | Factor Significance (%) | Differences between FS and NPV FS | Factor Significance (FA) After Considering NPV |
|---|---|---|---|---|---|---|---|
| F-D_01 | 15.38 | 0.154 | 0.154 | 0.0237 | 2.37 | 0.0003 | 0.0240 |
| F-D_02 | 57.69 | 0.577 | 0.577 | 0.3328 | 33.28 | 0.0083 | 0.3412 |
| F-D_03 | 7.69 | 0.077 | 0.154 | 0.0118 | 1.18 | 0.0004 | 0.0122 |
| F-D_04 | 5.00 | 0.050 | 0.115 | 0.0058 | 0.58 | 0.0001 | 0.0059 |
| F-D_05 | 12.69 | 0.127 | 0.269 | 0.0342 | 3.42 | 0.0002 | 0.0344 |
| F-D_06 | 14.23 | 0.142 | 0.192 | 0.0274 | 2.74 | 0.0003 | 0.0277 |
| F-D_07 | 15.38 | 0.154 | 0.423 | 0.0651 | 6.51 | 0.0016 | 0.0667 |
| F-D_08 | 10.00 | 0.100 | 0.231 | 0.0231 | 2.31 | 0.0004 | 0.0235 |
| F-D_09 | 1.92 | 0.019 | 0.038 | 0.0007 | 0.07 | 0.0000 | 0.0008 |
| F-D_10 | 5.00 | 0.050 | 0.154 | 0.0077 | 0.77 | 0.0001 | 0.0078 |
| F-P_01 | 42.31 | 0.423 | 0.423 | 0.1790 | 17.90 | 0.0034 | 0.1824 |
| F-P_02 | 27.69 | 0.277 | 0.577 | 0.1598 | 15.98 | 0.0010 | 0.1608 |
| F-P_03 | 36.54 | 0.365 | 0.385 | 0.1405 | 14.05 | 0.0018 | 0.1423 |
| F-P_04 | 21.15 | 0.212 | 0.231 | 0.0488 | 4.88 | 0.0006 | 0.0494 |
| F-P_05 | 38.08 | 0.381 | 0.615 | 0.2343 | 23.43 | 0.0029 | 0.2372 |
| F-P_06 | 19.23 | 0.192 | 0.192 | 0.0370 | 3.70 | 0.0007 | 0.0377 |
| F-R_01 | 9.62 | 0.096 | 0.346 | 0.0333 | 3.33 | 0.0004 | 0.0337 |
| F-R_02 | 3.08 | 0.031 | 0.115 | 0.0036 | 0.36 | 0.0002 | 0.0037 |
| F-R_03 | 57.69 | 0.577 | 0.615 | 0.3550 | 35.50 | 0.5769 | 0.9320 |
| F-R_04 | 61.54 | 0.615 | 0.615 | 0.3787 | 37.87 | 0.1420 | 0.5207 |
| F-R_05 | 65.38 | 0.654 | 0.654 | 0.4275 | 42.75 | 0.4810 | 0.9085 |
| F-R_06 | 9.62 | 0.096 | 0.115 | 0.0111 | 1.11 | 0.0050 | 0.0161 |
| F-R_07 | 7.69 | 0.077 | 0.077 | 0.0059 | 0.59 | 0.0036 | 0.0095 |
| F-R_08 | 34.62 | 0.346 | 0.346 | 0.1198 | 11.98 | 0.0180 | 0.1378 |
| F-R_09 | 38.46 | 0.385 | 0.692 | 0.2663 | 26.63 | 0.0799 | 0.3462 |
| F-R_10 | 16.54 | 0.165 | 0.385 | 0.0636 | 6.36 | 0.0016 | 0.0652 |

**Table 2.** *Cont.*

| Factor | Impact Intensity (%) | Impact Intensity | Probability | Factor Significance (FS) | Factor Significance (%) | Differences between FS and NPV FS | Factor Significance (FA) After Considering NPV |
|---|---|---|---|---|---|---|---|
| R-D_01 | 34.62 | 0.346 | 0.346 | 0.1198 | 11.98 | 0.0022 | 0.1221 |
| R-D_02 | 23.08 | 0.231 | 0.231 | 0.0533 | 5.33 | 0.0013 | 0.0546 |
| R-D_03 | 1.92 | 0.019 | 0.077 | 0.0015 | 0.15 | 0.0000 | 0.0015 |
| R-D_04 | 13.08 | 0.131 | 0.308 | 0.0402 | 4.02 | 0.0008 | 0.0410 |
| R-D_05 | 69.23 | 0.692 | 0.692 | 0.4793 | 47.93 | 0.0060 | 0.4853 |
| R-D_06 | 15.00 | 0.150 | 0.654 | 0.0981 | 9.81 | 0.0441 | 0.1422 |
| R-D_07 | 9.04 | 0.090 | 0.769 | 0.0695 | 6.95 | 0.0104 | 0.0800 |
| R-P_01 | 51.54 | 0.515 | 0.538 | 0.2775 | 27.75 | 0.0087 | 0.2862 |
| R-P_02 | 11.54 | 0.115 | 0.115 | 0.0133 | 1.33 | 0.0002 | 0.0135 |
| R-P_03 | 15.38 | 0.154 | 0.154 | 0.0237 | 2.37 | 0.0001 | 0.0238 |
| R-P_04 | 42.31 | 0.423 | 0.423 | 0.1790 | 17.90 | 0.0089 | 0.1879 |
| R-P_05 | 6.15 | 0.062 | 0.385 | 0.0237 | 2.37 | 0.0030 | 0.0266 |
| R-P_06 | 19.23 | 0.192 | 0.192 | 0.0370 | 3.70 | 0.0018 | 0.0388 |
| R-R_01 | 6.54 | 0.065 | 0.308 | 0.0201 | 2.01 | 0.0045 | 0.0246 |
| R-R_02 | 7.69 | 0.077 | 0.308 | 0.0237 | 2.37 | 0.0053 | 0.0290 |
| R-R_03 | 17.50 | 0.175 | 0.731 | 0.1279 | 12.79 | 0.0384 | 0.1663 |
| R-R_04 | 25.77 | 0.258 | 0.731 | 0.1883 | 18.83 | 0.0424 | 0.2307 |
| R-R_05 | 17.69 | 0.177 | 0.423 | 0.0749 | 7.49 | 0.0075 | 0.0823 |
| R-R_06 | 4.23 | 0.042 | 0.115 | 0.0049 | 0.49 | 0.0005 | 0.0054 |
| C-D_01 | 5.38 | 0.054 | 0.192 | 0.0104 | 1.04 | 0.0003 | 0.0106 |
| C-D_02 | 25.00 | 0.250 | 0.846 | 0.2115 | 21.15 | 0.0635 | 0.2750 |
| C-D_03 | 0.00 | 0.000 | 0.000 | 0.0000 | 0.00 | 0.0000 | 0.0000 |
| C-D_04 | 14.62 | 0.146 | 0.192 | 0.0281 | 2.81 | 0.0084 | 0.0365 |
| C-D_05 | 40.00 | 0.400 | 0.808 | 0.3231 | 32.31 | 0.2423 | 0.5654 |
| C-D_06 | 28.46 | 0.285 | 0.692 | 0.1970 | 19.70 | 0.0049 | 0.2020 |
| C-D_07 | 1.54 | 0.015 | 0.077 | 0.0012 | 0.12 | 0.0000 | 0.0012 |
| C-P_01 | 6.73 | 0.067 | 0.346 | 0.0233 | 2.33 | 0.0004 | 0.0237 |
| C-P_02 | 9.62 | 0.096 | 0.615 | 0.0592 | 5.92 | 0.0089 | 0.0680 |
| C-P_03 | 13.65 | 0.137 | 0.885 | 0.1208 | 12.08 | 0.0181 | 0.1389 |
| C-P_04 | 12.88 | 0.129 | 0.885 | 0.1140 | 11.40 | 0.0057 | 0.1197 |
| C-P_05 | 20.77 | 0.208 | 0.846 | 0.1757 | 17.57 | 0.0066 | 0.1823 |
| C-P_06 | 3.46 | 0.035 | 0.154 | 0.0053 | 0.53 | 0.0001 | 0.0054 |
| C-P_07 | 19.81 | 0.198 | 0.654 | 0.1295 | 12.95 | 0.0016 | 0.1311 |
| C-R_01 | 3.08 | 0.031 | 0.038 | 0.0012 | 0.12 | 0.0001 | 0.0012 |
| C-R_02 | 8.08 | 0.081 | 0.231 | 0.0186 | 1.86 | 0.0003 | 0.0190 |
| C-R_03 | 13.08 | 0.131 | 0.462 | 0.0604 | 6.04 | 0.0008 | 0.0611 |
| C-R_04 | 5.00 | 0.050 | 0.192 | 0.0096 | 0.96 | 0.0001 | 0.0097 |
| C-R_05 | 3.46 | 0.035 | 0.154 | 0.0053 | 0.53 | 0.0001 | 0.0054 |
| C-R_06 | 7.12 | 0.071 | 0.308 | 0.0219 | 2.19 | 0.0003 | 0.0222 |

These results point to the possibilities of risk management and quantifying risks and impacts on the technical and financial aspects, in the form of efficiency. The factor significance (FS) confirms the impact of technical risks. The advantage of this methodology and these indicators are based on the fact that they are related to the construction projects in question. On this basis, it is possible to prioritize risk management on selected processes that are statistically impacted the most heavily on the technical side.

From the point of view of efficiency, it is essential not only to perceive this aspect in terms of the impact on costs in absolute terms at a given time but also to consider the influence of time on selected technological processes and the time value of the necessary investment. This predominantly works based on obtaining an accurate coefficient for the value taken into account and is also based on the time perspective. In other words, the endeavor is about respecting the additional costs and time needed to eliminate the

risk. Each process within the realization of concrete constructions has a technologically different duration, which has different values within the FS coefficient after considering the net present value. These values are essential from the point of view of efficiency. The most significant difference between the coefficients was for the C-D_05 process, which represents the quality control of the concrete process. On the contrary, it caused the most negligible differences in processes, such as cold joint compliance or the method used for the compaction process.

The time aspect of efficiency is not negligible in time-consuming processes. Therefore, the calculation method's design and its subsequent implementation on other projects can afford a high degree of support for the decision-making and prioritization of risk management for selected construction processes. However, when comparing results from other studies, it is not easy to reach a conclusion and compare the results. To date, all research results have been mainly focused on factors that influence or quantify risk. However, there needs to be more research on where it would be possible to propose a methodology for quantifying impacts and a tool that would support the decision-making and prioritization of risk management in selected concrete works. Therefore, the results of this research can significantly help with this issue in practice.

## 4. Conclusions

The issue of risk management in selected construction projects is significant, which is also confirmed by the results of this research. Prioritizing processes that significantly affect the technical side of a construction project and its efficiency is a significant step forward from the point of view of the management of a construction project. Research to date has pointed to factors that affect the risks of construction projects.

The issue of risk management, therefore, poses several challenges from the point of view of practice. The most high-impact risks were identified within the formwork group, mainly in connection with formwork preparation processes. Conversely, the most risks with a high probability were detected in the concrete group, especially within the transport and preparation processes. Contractors should, therefore, devote an increased level of quality control to these processes.

From an economic point of view, or an efficiency point of view, it is essential not only to perceive this aspect in the impact on costs in absolute terms at a given time but also to consider the influence of time on selected technological processes and the time value of the necessary investment. From an economic point of view, the key processes are documentation, compliance with the project, and the availability of a release agent. This has the most impact considering the net present value factor, which is key from an efficiency point of view.

The most significant need is to effectively manage the risk, in other words, to prioritize and focus on those risks that represent the most significant threat and the probability of their fulfillment. From the point of view of effective risk management, it is crucial to focus on those risks that have the most significant impact on the technical side (such as the difficulty of processes, the time needed for implementation, and impacts on other subsequent processes) and the financial side, that is, on the efficiency side. To a great extent, this has the effect of assessing the level of risk and impacts based on the potential investment, which also considers the time value. To achieve the effective management of construction projects and their risks, this means focusing not only on those risks that have the most significant impact on the complexity of the processes and those that are the most important from the point of view of the scope of investments but also affect the real value in the form of costs.

This research aimed to analyze and verify the degree of impact of individual factors on construction efficiency when building concrete structures of the selected type of buildings. Furthermore, the impact of risk management on the efficiency of the construction project of residential buildings was analyzed and the method of quantifying the impact on the technical and financial aspects, based on the calculated coefficients, was proposed.

This research pointed to the specific risks and offered a model for quantifying risks and impacts, representing a risk management tool for the listed buildings. This approach can be applied to risk management. As for the next direction of research, this should be aimed at expanding the types of construction projects beyond those that were the subject of this research. Likewise, the research results are based on research projects focused on construction projects in Central Europe. Other construction and weather conditions in the southern or other climate zones may change some coefficients.

**Author Contributions:** Conceptualization, M.Š., T.M. and M.S.; methodology, M.Š., T.M. and M.S.; software, M.Š., T.M. and M.S.; validation, M.Š., T.M. and M.S.; formal analysis, M.Š., T.M. and M.S.; investigation, M.Š., T.M., M.S. and D.V.; resources, M.Š., T.M. and M.S.; data curation, M.Š., T.M. and M.S.; writing—original draft preparation, M.Š., T.M. and M.S; writing—review and editing, M.Š., T.M. and M.S.; visualization, M.Š., T.M. and M.S.; supervision, M.Š., T.M. and M.S.; project administration, M.Š., T.M. and M.S.; funding acquisition, M.Š., T.M. and M.S. All authors have read and agreed to the published version of the manuscript.

**Funding:** The paper was not funded by any sources.

**Informed Consent Statement:** Not applicable.

**Data Availability Statement:** Not applicable.

**Acknowledgments:** The paper presents the partial research results of the following projects: VEGA 1/0336/22, research on the effects of lean production/lean construction methods on increasing the efficiency of on-site and off-site construction technologies; project KEGA 009TUKE-4/2022, an interactive tool for designing a safe construction site in an immersive virtual reality; and project APVV-17-0549, research into knowledge-based and virtual technologies for intelligent design and the realization of building projects, with an emphasis on economic efficiency and sustainability.

**Conflicts of Interest:** The authors declare no conflict of interest.

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
