# Peer review of "Risk Management on Concrete Structures as a Tool for the Control of Construction Efficiency"

_sustainability, doi:10.3390/su15129577_

Round 1

Reviewer 1 Report

Review report. Minor corrections.

The manuscript “Risk Management on Concrete Structures as a Tool for Control of Construction Efficiency” analyzes impact of risk management on the efficiency of the construction project of residential buildings and proposes a method of quantifying the impact on the technical and economic side based on the calculated coefficients. The research results provide coefficients for determining the technical and economic impacts that can be implemented for risk management on Concrete Structures of construction projects of residential buildings. The research is based on data from residential building projects in Central Europe.

The manuscript offers very interesting theme. The results are useful for actual construction practice. Research is scientifically sound and novel.

The manuscript is clear, relevant for the field and presented in a well-structured manner.

Before the manuscript is accepted for publication, it has to undergo few changes and corrections. The list of specific comments is provided below:

1.       Abstract is to long. It should be abridged to 200-300 words.

2.       Keywords should not repeat the terms already stated in the title.

3.       The Introduction is adequately written. The provided review of literature is clear, comprehensive and of relevance to the field. Authors identified the gap in knowledge and highlighted it in the last paragraph. 

4.       Materials & Methods chapter should be renamed into ‘Methods’ or ‘Methods applied’ or something similar.

5.        Result & Discussion chapter gives a clear presentation of results and models. Everything seems scientifically sound. The experimental design is appropriate to test the hypothesis. There are no errors of fact and logic.

6.       Figures and tables are appropriate.

7.       The conclusions are adequate.  They follow the findings highlighted in Results & discussion chapter.  

8.       The cited references are appropriate and up-to-date (within the last 5 years or so). Used literature is relevant. There is no excessive number of self-citations.

9.       The text should be read and corrected by English language native speaker or professional proof-reader.

There are typing mistakes and misused phrases. The text should be read and corrected by English language native speaker or professional proof-reader.

Author Response

Dear Reviewer,

First of all, on behalf of the entire author team, let me thank you very much for the valuable comments and suggestions to improve our manuscript. The document was extensively revised based on the reviewers' comments and a lot of information was added to clarify possible questions and ambiguities.

We have highlighted all modifications related to comments in the modified document. We have honestly processed all comments and modified them according to your proposal. The MDPI English editing process is currently underway for clearer readability and correctness of the document.

Once again, thank you very much for your comment and suggestions for improvement. We really appreciate it.

Reviewer 2 Report

Dear Authors,

Please explain why and how you have chosen your projects based on which you have collected your research data.

What were the questions set in your questionnaire or interview based on which you have processed their answers?

Please explain more in detail your research process, as it is not clear at all.

Author Response

Dear Reviewer,

First of all, on behalf of the entire author team, let me thank you very much for the valuable comments and suggestions to improve our manuscript. The document was extensively revised based on the reviewers' comments and a lot of information was added to clarify possible questions and ambiguities.

We have highlighted all modifications related to comments in the modified document. We have honestly processed all comments and modified them according to your proposal. The MDPI English editing process was done.

Some of important part for methodology extend. We explain asked parts in the chapter 2.3, where is described selection and reason for selecting the data source as well as that the questions were set up according to processes listed in Table 1. In this part of paper is also explained whole research process, in details.

Once again, thank you very much for your comment and suggestions for improvement. We really appreciate it.

Reviewer 3 Report

Please find the attachment

Need to be modified.

Author Response

Dear Reviewer,

First of all, on behalf of the entire author team, let me thank you very much for the valuable comments and suggestions to improve our manuscript. The document was extensively revised based on the reviewers' comments and a lot of information was added to clarify possible questions and ambiguities.

We have highlighted all modifications related to comments in the modified document. We have honestly processed all comments and modified them according to your proposal. The MDPI English editing process was done. Submitted document is after MDPI English editing process.

Some of important part for methodology extend. We explain asked parts in the chapter 2.3, where is described selection and reason for selecting the data source as well as that the questions were set up according to processes listed in Table 1. In this part of paper is also explained whole research process, in details.

Relevant sentence was modified to be clear much more.

Once again, thank you very much for your comment and suggestions for improvement. We really appreciate it.

Round 2

Reviewer 3 Report

Please find the attachment

English language can be improved. 

Author Response

Dear Editor,

Thank you very much for your opinion and valuable comments. We accept all your comments, and based on your suggestions, we have modified the attached document. In addition to the mentioned content changes and formal ones (all highlighted in grey), the MDPI English editor did language proofreading.

Thanks again for the opportunity to improve this manuscript.
